



# Role of air-sea fluxes and ocean surface density on the production of deep waters in the eastern subpolar gyre of the North Atlantic

Tillys Petit[1], M. Susan Lozier[1], Simon A. Josey[2], and Stuart A. Cunningham[3]

[1]School of Earth and Atmospheric Sciences, Georgia Institute of Technology, Atlanta, GA, USA
[2]National Oceanography Centre, Southampton, UK
[3]Scottish Association for Marine Science, Oban, UK

*Correspondence to*: Tillys Petit (tillys.petit@gatech.edu)

**Abstract.** Wintertime convection in the North Atlantic Ocean is a key component of the global climate as it produces dense waters at high latitudes that flow equatorward as part of the Atlantic Meridional Overturning Circulation (AMOC). Recent work has highlighted the dominant role of the Irminger and Iceland basins in the production of North Atlantic Deep Water. Dense water formation in these basins is mainly explained by buoyancy forcing that transforms surface waters to the deep waters of the AMOC lower limb. Air-sea fluxes and the ocean surface density field are both key determinants of the buoyancy-driven transformation. We analyze these contributions to the transformation in order to better understand the connection between atmospheric forcing and the AMOC. More precisely, we study the impact of air-sea fluxes and the ocean surface density field on the transformation of subpolar mode water (SPMW) in the Iceland Basin, a water mass that 'pre-conditions' dense water formation downstream. Analyses using 40 years of observations (1980–2019) reveal that the variance in SPMW transformation is mainly influenced by the variance in density at the ocean surface. This surface density is set by a combination of advection, wind-driven upwelling and surface fluxes, the latter explaining ~30% of the variance in outcrop area as expressed by the surface area between the outcropped SPMW isopycnals. The key role of the surface density on SPMW transformation partly explains the unusually large SPMW transformation in winter 2014–2015 over the Iceland Basin.



# 1 Introduction and Background

Recent observational studies have identified two main source regions for the dense waters that constitute the lower limb of the Atlantic Meridional Overturning Circulation (AMOC): the Nordic Seas, and the Irminger and Iceland basins (Bringedal et al., 2018; Chafik & Rossby, 2019; Lozier et al., 2019; Petit et al., 2020). While these two regions produce approximately the same amount of dense waters, the interannual variability of the AMOC in the subpolar North Atlantic can be largely attributed to variability in the dense waters produced in the eastern subpolar region, rather than those imported from the Nordic Seas. A recent study based on observations shows that the production of dense waters in the Irminger and Iceland basins is mainly due to local buoyancy forcing (Petit et al., 2020). Specifically, between the Greenland-Scotland Ridge and the OSNAP East section (Fig. 1), 7.0 ± 2.5 Sv of light water was transformed into dense water of the AMOC lower limb in 2014–2016. This transformation is consistent with an average overturning of 6.6–7.6 ± 3.8 Sv between the two sections during those same years.

Because the transformation of dense waters mainly drives AMOC variability in this region, we seek to understand what factors drive variability in the transformation, which was twice as large in winter 2014–2015 as it was in winter 2015–2016 (Petit et al., 2020). Two variables are key to transformation estimates: the air-sea fluxes and the ocean surface densities. The air-sea fluxes are known to be highly variable in time over the subpolar gyre. For instance, Josey et al. (2019) showed strong seasonal and interanual variability of the net heat flux over the Irminger Sea from a high-resolution surface flux mooring. Changes in net heat flux—as opposed to changes in the surface salinity field or in the surface freshwater flux—are considered to be mainly responsible for the amount of transformed water (Grist et al., 2014). Another recent study, however, has suggested that ocean surface properties play a role in the transformation of surface waters. Oltmanns et al. (2018) revealed that warm and fresh summers in the subpolar gyre are associated with a reduced buoyancy loss the following winter. Hydrographic property changes are thus expected to affect the transformation of surface water by modifying the surface density fields.



Observational studies in the last few years have highlighted large temperature and salinity changes over the eastern subpolar gyre (Holliday et al., 2020; Josey et al., 2018). The density anomalies associated with these property changes are expected to influence the overturning over the Irminger and Iceland Basin, and thus the amount of dense water transported southward to the subtropical gyre (Jackson et al., 2016; Zou et al., 2020). At odds with this expectation, however, is a recent study (Fu et al., 2020) that showed a relatively stable AMOC state since the 1990s, in spite of large hydrographic property changes over the subpolar gyre during the same time period. This stability suggests either that hydrographic property changes have only a small influence on the transformation or that a lagged response of the overturning to property changes applies over longer time scales than their 30-years period of observations. Motivated by these past studies, we assess the role of hydrographic property changes on the formation of dense water in the eastern subpolar North Atlantic.

In this study, we focus on the factors that determine the transformation of subpolar mode water (SPMW) in the Iceland Basin. SPMW is identified by uniform hydrographic properties in the winter mixed layer of the subpolar gyre (McCartney & Talley, 1982) and is often defined by potential vorticity < $4 \times 10^{-11}$ m$^{-1}$s$^{-1}$ over the eastern subpolar gyre (Talley, 1999). In contact with the atmosphere, its hydrographic properties vary along the cyclonic pathway of the subpolar gyre, increasing in density westward across the Iceland Basin from 27.3 to 27.5 kg m$^{-3}$ (Brambilla & Talley, 2008; Thierry et al., 2008). The SPMW, considered the 'pre-conditioned' water mass that forms North Atlantic Deep Water (NADW), is further densified in the Irminger and Labrador Seas (de Boisséson et al., 2012; Brambilla et al., 2008).

The westward freshening and cooling of SPMW, evident in the OSNAP East hydrographic section from 2014–2016 (Fig. 1b), transforms light SPMW in the North Atlantic Current (NAC) branches to the relatively dense NADW in the East Greenland Current. As observed by the spread of temperatures and salinities along the 27.74 kg m$^{-3}$ in the Irminger Gyre (Fig. 1b), the NADW imported from the Labrador Sea to the Irminger Sea has a similar density to (Faure & Speer, 2005; Fischer et al., 2018; Jakobsen et al., 2003; Petit et al., 2019), but is slightly fresher and colder than, the NADW formed by SPMW densification within the eastern subpolar basin and exported via the East Greenland Current. The



transfromation of SPMW in the layer 27.3–27.5 kg m$^{-3}$ over the Iceland Basin (line segments 1–4 in Figure 1a) is the focus of our study.

As explained in Sect. 2, we use observational datasets and sensitivity experiments to explore the
85  dependence of SPMW transformation on the air-sea fluxes and surface densities over a 40-year period. We examine the linkage between buoyancy forcing and the surface density field in Sect. 3, while the sensitivity experiments are discussed in Sect. 4. Section 5 is focused on the unusually large SPMW transformation observed in winter 2014–15 and we summarize our results in Sect. 6.

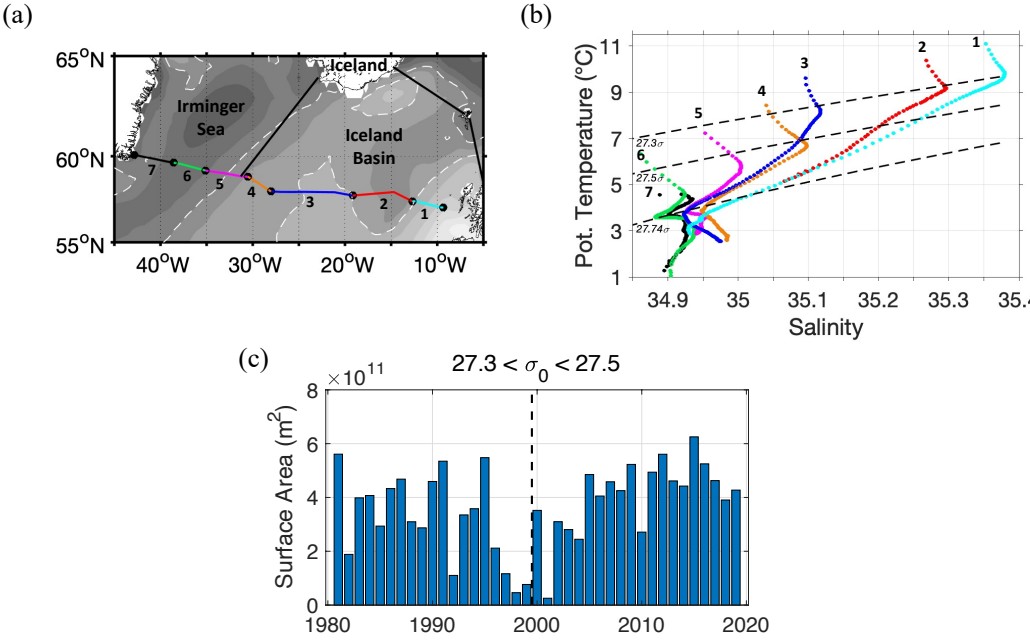

90

**Figure 1.** (a) Sea surface density (kg m$^{-3}$) averaged over winter (December to April) 1980–2019; contour interval is 0.1 kg m$^{-3}$. White dashed lines outline the isopycnals 27.3 and 27.5 kg m$^{-3}$. The OSNAP East section, divided into 7 coloured subsections, forms the southern boundary of closed domain for our study. (b) Potential temperature/salinity diagram at OSNAP East averaged over 2014–2018 for (from east to west) the three northward branches of the North Atlantic Current (NAC) in the Iceland Basin (1–3); southward East Reykjanes
95  Ridge Current (4); northward Irminger Current (5); northward Irminger Gyre (6) and southward East Greenland Current (7). Dashed black lines show the potential densities $\sigma_0$ = 27.3, 27.5 and 27.74 kg m$^{-3}$. (c) Surface area (m$^2$) over the Iceland Basin in January.




## 2 Materials and Methods

### 2.1 The OSNAP observations

We used gridded data across the OSNAP East section to observe the evolution of hydrographic properties
in the Iceland Basin and Irminger Sea (Fig. 1; Lozier et al. (2017)). Based on thirty-two moorings, the
gridded section has a nominal horizontal resolution of ~25 km from the Scottish shelf to the southeast tip
of Greenland, and a vertical resolution of ~20 m (Lozier et al., 2019). The data are composed of
continuous measurements of salinity, temperature and velocity between August 2014 to May 2018. The
gridded data set was constructed by combining OSNAP mooring data with other observations in the
region following an Objective Analysis (Li et al., 2017). The other observations include Argo Profiling
Floats, OSNAP gliders, shipboard Conductivity, Temperature and Depth stations carried out during the
2014 and 2016 OSNAP cruises, and the WORLD Ocean Atlas 2018 climatology.

### 2.2 Transformation estimated from Atmospheric Reanalysis

To estimate the transformation of surface water by air-sea fluxes, we use heat and freshwater fluxes of
the European Centre for Medium Range Weather Forecasts Reanalysis 5 (ERA5) atmospheric reanalysis
(Poli et al., 2016). These monthly fields are combined with monthly surface temperature and salinity
fields over the eastern subpolar gyre between 1980–2019. Surface temperature is derived from ERA5,
while salinity, at 5-m depth, is derived from quality-controlled profiles in EN4.2.1 (Good et al., 2013).
With a horizontal resolution of 1°, the gridded salinity field is subsampled onto the finer ERA5 horizontal
grid of 30 km. We also use EN4.2.1 to calculate potential vorticity ($\frac{-f}{\rho} \frac{\partial \rho}{\partial z}$) in the Iceland Basin, where f
is the Coriolis parameter and ρ is the potential density.

The evaporation (*E*), precipitation (*P*), and net heat flux into the ocean (*Q*) are used to estimate the surface-
forced transformation across an isopycnal, σ, according to (Speer & Tziperman, 1992; Tziperman, 1986;
Walin, 1982):

$$\text{SFT}(\sigma^*) = \frac{1}{\Delta\sigma} \iint \left[ -\frac{\alpha}{C_P} Q + \beta \frac{S}{1-S}(E-P) \right] \Pi(\sigma)\, dx\, dy$$



where,

$$\Pi(\sigma) = \begin{cases} 1 & \text{for } |\sigma - \sigma^*| \leq \frac{\Delta\sigma}{2} \\ 0 & \text{elsewhere} \end{cases}$$

α is the thermal expansion coefficient, β is the haline contraction coefficient, $C_P$ is the specific heat, and

S is the 5-m salinity. Temperature and salinity are used to calculate surface density (Gill, 1982). For each month and each isopycnal, σ, the local buoyancy flux (term in square brackets) is integrated over the surface area of the associated density bin, Δσ. Thus, SFT has a non-zero value for those months when the specified isopycnals outcrop, otherwise it is set to zero. To focus on the transformation within the SPMW layer (27.3–27.5 kg m⁻³), we estimate the transformation across the isopycnal 27.4 kg m⁻³ toward higher

or lower densities using a density bin size of $\Delta\sigma = 0.2$ kg m⁻³. From this equation, we note that air-sea fluxes influence the transformation both directly, through the buoyancy flux term, and indirectly by partially setting the area covered by a particular surface density range, which is also influenced by the ocean circulation. We thus expect these two factors (i.e. surface density and buoyancy flux) to share some dependence. We explore this dependence in Sect. 3.


Previous studies have shown the utility of transformation maps to highlight the migration of intense transformation for different density bins over the subpolar and subtropical gyres (Brambilla et al., 2008; Maze et al., 2009). Although the equation above can be used to integrate the transformation for any density bin, the integrand is calculated only over the surface area for the specified isopycnals that bracket

the density of the source water for SPMW. Thus, the integration yields the magnitude of the transformation over the domain. In Sect. 4, we visualize the spatial distribution of the SPMW transformation variability by mapping the standard deviation of SFT and explore its sensitivity to two factors: the surface density and the air-sea fluxes. As further described below, annual SPMW transformations are estimated with time variable and/or monthly climatological surface density ($\mathcal{D}$, which

includes the variability of surface salinity and temperature) and air-sea fluxes ($\mathcal{F}$, which includes the variability of $Q$, $E$ and $P$).



## 2.3 Surface area for the source water

The spatial pattern of the climatological mean surface density shows that the SPMW layer outcrops over a large part of the Iceland Basin during winter (Fig. 1a). To quantify the interannual variability of this
outcrop between winters, we estimate the area between the two outcropped isopycnals of the SPMW layer (27.3–27.5 kg m$^{-3}$) over our study domain, which is bounded by the OSNAP East section (1–4), the top of the Reykjanes Ridge and the Iceland–Faroe–Scotland Ridge, as indicated in Fig. 1a. This area is hereafter referred to as the surface area of the source water, or simply the surface area.

The mean surface area of the source water over the Iceland Basin is 3.67x10$^{11}$ m$^2$ and is, surprisingly, highly variable over the period 1980–2019, with a standard deviation of 1.53x10$^{11}$ m$^2$ (Fig. 1c). Interestingly, the surface area is more variable between 1980–1999 (1.63x10$^{11}$ m$^2$) than between 2000–2019 (1.34x10$^{11}$ m$^2$). The impact of this difference on SPMW transformation will be investigated in Sect. 4.

## 2.4 Gyre Boundary estimated from AVISO

To estimate the subpolar gyre boundary, we use monthly absolute dynamic topography fields from the gridded ¼° AVISO (Archiving, Validation and Interpretation of Satellite Oceanographic data center) altimeter products distributed by CMEMS (Copernicus Marine Environment Monitoring Service). For ease, absolute dynamic topography is hereafter referred to as sea surface height (SSH). Following
previous work, the gyre boundary is defined as the largest closed contour of the monthly SSH field with 1-cm contour intervals (Foukal & Lozier, 2017).

## 3 Influence of buoyancy loss on the structure of the upper ocean density

We begin our study of the influence of air-sea fluxes and ocean surface densities on the SPMW transformation in the Iceland Basin by evaluating the independence of these variables. We first examine
the influence of the local buoyancy flux on the vertical density profile for each winter over the OSNAP period from 2014 to 2018, a period during which Petit et al. (2020) identified large interannual variability in the buoyancy forcing. These profiles reveal the ventilation of SPMW (again, identified by potential





vorticity $< 4 \times 10^{-11}$ m$^{-1}$s$^{-1}$) at the beginning of winter and its restratification at the onset of spring (Fig. 2a). The 2014–2015 winter stands out among these profiles, as it is marked by both the strongest buoyancy

flux ($8 \times 10^{-6}$ W m$^{-2}$ in December; Fig. 2a) and the deepest SPMW (600 m in March; Fig. 2b). Conversely, the 2016–2017 winter is associated with a weak buoyancy flux ($4 \times 10^{-6}$ W m$^{-2}$ in December) and a shallow SPMW (250 m in March). Despite these agreements, the linkage between the SPMW thickness and the buoyancy flux over the period 1999-2019 is weak, with a correlation of only 0.52 (Fig. 2c). It is possible that this lower than expected correlation can be explained by year to year variability in the surface density

field. Even if the buoyancy forcing is relatively strong over the Iceland Basin for a given winter, the surface area needs to be large enough to have a sizeable impact on the SPMW layer. This possibility raises the question as to the influence of the buoyancy forcing on the surface area.

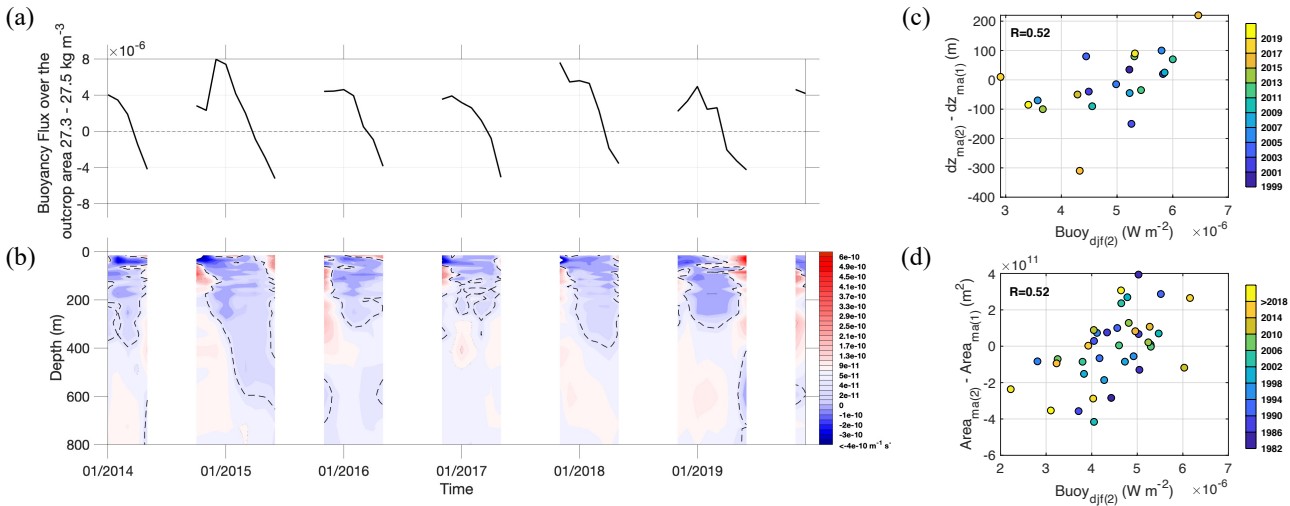

**Figure 2.** (a) Buoyancy flux (W m$^{-2}$) and (b) potential vorticity (m$^{-1}$ s$^{-1}$) averaged over the source density area in the Iceland Basin. The isopycnals 27.3 and 27.5 kg m$^{-3}$ do not outcrop over the Iceland Basin during summer. The dashed black line shows the potential vorticity $4 \times 10^{-11}$ m$^{-1}$ s$^{-1}$. (c) Correlations between the buoyancy flux averaged over December-January-February in the second winter (Buoy$_{djf(2)}$), and the difference in SPMW thickness March-April between the end of the first (dz$_{ma(1)}$) and second (dz$_{ma(2)}$) winters over 1999–2019. The SPMW thickness was estimated by the depth of the potential vorticity isoline $4 \times 10^{-11}$ m$^{-1}$ s$^{-1}$ over the source water area (as shown in Fig.

2b). EN4.2.1 data before 1999 are not used because of large uncertainties due to scarce observations at depth. (d) Correlation between the buoyancy flux averaged over December-January-February in the second winter (Buoy$_{djf(2)}$), and the difference in surface area between the end of the first (area$_{ma(1)}$) and second winters (area$_{ma(2)}$).





To answer this question, we subtract the surface area at the end of each winter from the surface area at the end of the following winter, for each year. We compare this area difference to the buoyancy forcing over the second winter as we want to understand whether the buoyancy forcing is setting the surface area observed at the end of that second winter (Fig. 2d). Though a strengthening of the buoyancy forcing generally leads to an expansion of the surface area, the buoyancy flux in a given winter can explain only 30% of the surface density change in the Iceland Basin. The remaining variance can be attributed to ocean advection, mixing and wind-driven upwelling, although no single process is likely dominant. To conclude, we assume that the dependence of the surface density field on the air-sea fluxes is sufficiently small so that we can separately explore the contributions of the air-sea fluxes and ocean surface densities on the interannual variability of SPMW transformation.

## 4 Influence of buoyancy loss on the structure of the upper ocean density

Our analysis of the correlation of buoyancy flux and surface area with SPMW transformation for the winters of 1980–2019 (Figures 3a and 3b) reveals that both variables have strong positive correlations with SPMW transformation over the Iceland Basin (0.70 and 0.67, respectively). Interpretation of these results is not straightforward because, as noted in Sect. 3, the surface area is dependent to some extent on the buoyancy flux ($R = 0.52$) as well as ocean advection, mixing and wind-driven upwelling. Thus, the correlation of SPMW transformation with the surface area in Fig. 3a reflects the combined effects of these four terms on the surface area within the SPMW density range. In turn, the correlation of SPMW transformation with the buoyancy flux in Fig. 3b reflects both direct (via the buoyancy flux term in the transformation equation, Sect. 2.2) and indirect (via modification of the surface area driven by the buoyancy flux) influences of the buoyancy flux on the SPMW transformation. These direct and indirect influences of the buoyancy flux together account for about half of the SPMW transformation variability. While these correlations do not enable us to establish causality, they do provide useful upper limits for the influence of different factors on transformation variability. Specifically, changes in surface area of the SPMW density range can explain at most 45% of the year-on-year transformation variability. Likewise, buoyancy fluxes account for up to 49%.




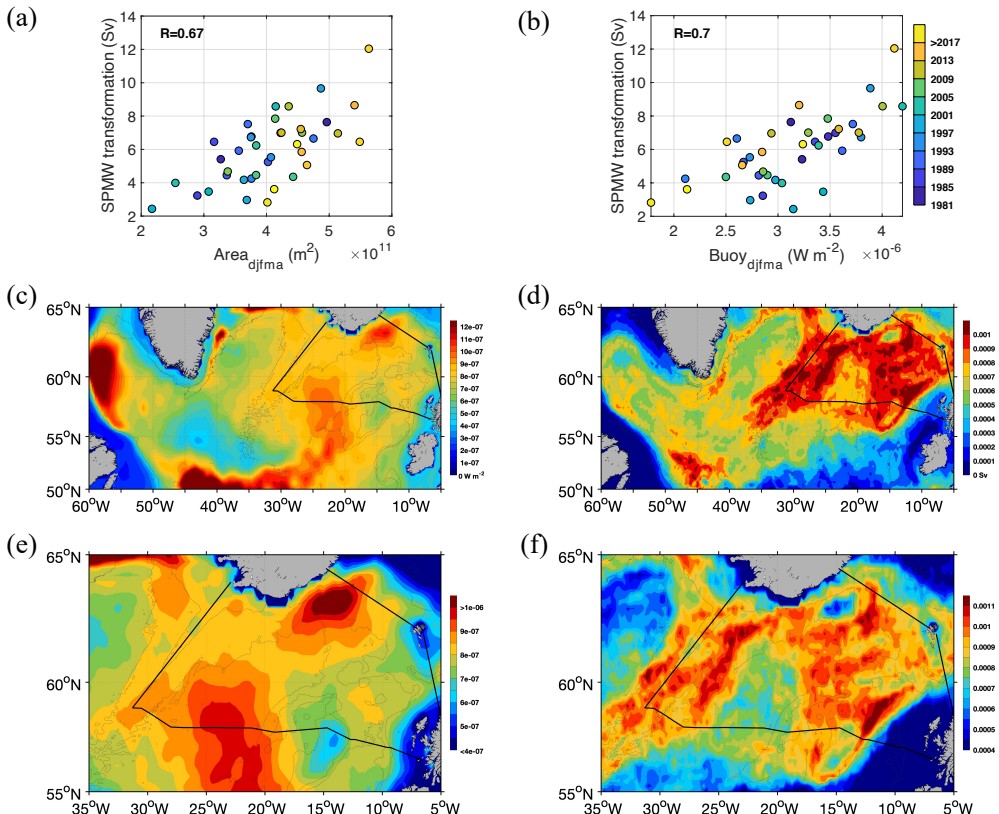

**Figure 3.** Correlations between the SPMW transformation to densities higher than 27.4 kg m$^{-3}$ in the Iceland Basin and the (a) buoyancy flux over the surface area 27.3–27.5 kg m$^{-3}$ in the Iceland Basin, and (b) the surface area 27.3–27.5 kg m$^{-3}$ in the Iceland Basin; all winter values (December to April). (c-e) Variance in buoyancy flux in winter. (d-f) Distribution of the interannual variability in SPMW transformation. The black boxes define the Iceland Basin area.

However, the spatial pattern of the buoyancy flux variance differs considerably from the spatial pattern of the SPMW transformation variance. Over the subpolar gyre, the largest variance in SPMW transformation is localized over the Reykjanes Ridge, Hatton-Rockall Plateau and Rockall Trough (Fig. 3d), while the largest variance in buoyancy flux is localized over the Labrador Sea and the NAC (Fig. 3c). Over the area defined by our black box, large variance in buoyancy flux is noted over the interior of the Iceland Basin (Fig. 3e), while the variance in SPMW transformation is small (Fig. 3f). These large spatial differences suggest that buoyancy fluxes are unlikely to be the only driver of variability in SPMW transformation.



We use sensitivity observation-based experiments to understand what is driving the spatial pattern in SPMW transformation. In Fig. 4, we compare the spatial pattern of the variance in SPMW transformation, SFT, from three experiments using the mean surface density $<\mathcal{D}>$, variable surface density ($\mathcal{D}'$), mean heat and freshwater fluxes $<\mathcal{F}>$ and variable fluxes $\mathcal{F}'$. In experiment (1), yearly transformations are

estimated as $\mathcal{D}'\mathcal{F}'$. In experiment (2), yearly transformations are estimated as $\mathcal{F}'<\mathcal{D}>$. And in experiment (3), yearly transformations are estimated as $\mathcal{D}'<\mathcal{F}>$. Surprisingly, the small variance in SPMW transformation of experiment (2) ($\mathcal{F}'<\mathcal{D}>$) compared to the two other experiments reveals that the variance in SPMW transformation is primarily driven by the variance in surface density. Similar results are obtained for the transformation of surface waters to a density $> 27.55\ \mathrm{kg\ m^{-3}}$, which is the

averaged isopycnal of the maximum AMOC at OSNAP East (Fig. 4, lower row). As expected, the region of large variance in transformation is shifted westward for this denser water mass, illustrating the progressive densification around the cyclonic pathways of the gyre.

The same analysis has been performed over the two periods differentiated by their standard deviation in

surface areas in Fig. 1c (1980–1999 and 2000–2019; Fig. 5). The interannual variability of SPMW transformation is driven primarily by the variance in surface density during both periods. This comparison also shows a relatively stronger variance of the transformation in 1980–1999 compared to 2000–2019, although their maxima remain localized over the Reykjanes Ridge and Rockall Trough. This highly variable transformation during the first period is consistent with a relatively large variation of surface area

during those years (Fig. 1c).




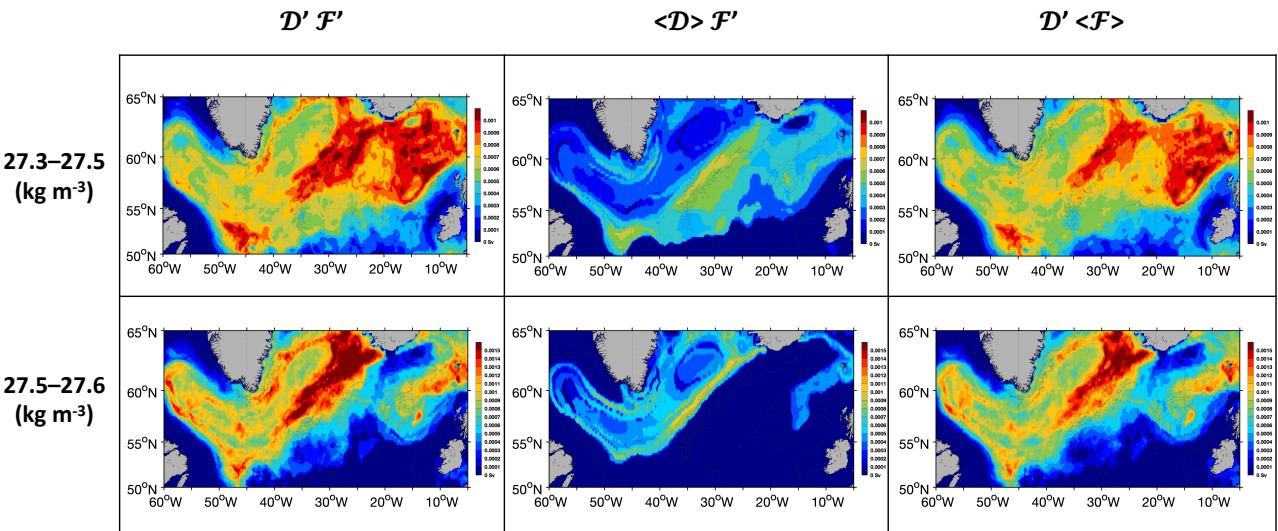

**Figure 4.** Sensitivity experiments of the variance in transformation estimated as the standard deviation of the yearly transformation integrated over the density range (upper row) 27.3–27.5 kg m$^{-3}$ and (lower row) 27.5–27.6 kg m$^{-3}$ layers during 1980–2019. The left column shows the variance in transformation estimated from monthly variable surface density ($\mathcal{D}$') and air-sea fluxes ($\mathcal{F}$'); the middle column shows the variance in transformation estimated from monthly variable heat and freshwater fluxes ($\mathcal{F}$') and climatological surface density (<$\mathcal{D}$>); the right column shows the variance in transformation estimated from monthly variable surface density ($\mathcal{D}$') and climatological heat and freshwater fluxes (<$\mathcal{F}$>). Variable and climatological density at surface stand for variable and climatological salinity and temperature at surface.

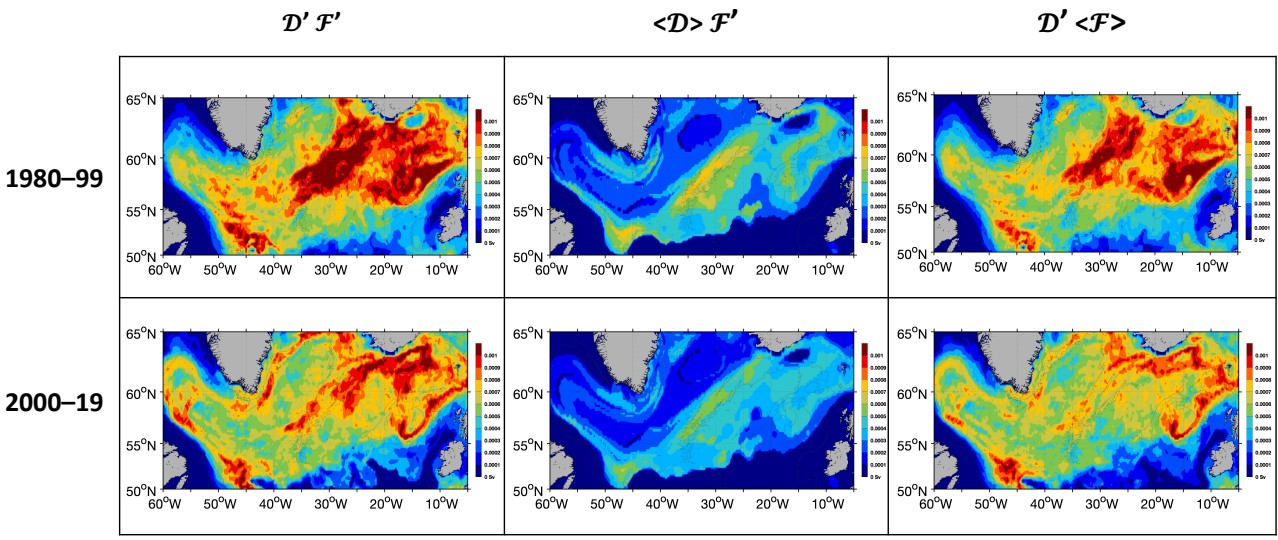

**Figure 5.** Same as in Fig. 4 for the density range 27.3–27.5 kg m$^{-3}$, but integrated over different periods of time.



## 5 Unusually large SPMW transformation in winter 2014–15

We next examine the conditions leading to intense transformation of SPMW over the Iceland Basin. The monthly time series of SPMW transformation in Fig. 6a clearly shows that the transformation of surface
water to a density greater than 27.4 kg m$^{-3}$ reached its largest value since 1980 during the 2014–15 winter, with a net transformation of 23.9 Sv in January 2015. Though the buoyancy anomaly is relatively strong during this winter, it is not the largest anomaly observed over the 40-year record (Fig. 6b), suggesting that additional conditions are required for large transformation. An examination of winter 2010–11 reveals that in spite of sizeable buoyancy forcing, the transformation appears to be limited by a relatively
small surface area. Winter 2013-14 shows the opposite: though the surface area is sizeable the relatively weak buoyancy forcing appears to limit the transformation. In contrast to these two winters, the large SPMW transformation in winter 2014–15 is associated with relatively strong buoyancy fluxes and a relatively large surface area that covers most of the Iceland Basin as defined by the black box in Fig. 6c.

By comparing the surface areas of the three winters (Fig. 6c), we note that their contraction or expansion is mainly related to the position of the isopycnal 27.3 kg m$^{-3}$ over the southeastern part of the Iceland Basin, south of our box. To characterize the spatial pattern of the surface area during winters of 'small' and 'large' areas, we select the five winters associated with the smallest, and five associated with the largest, surface areas of the source water (Fig. 7a). These areas were estimated over a larger box (shown
in Fig. 7c) so as to include the shift of the isopycnal 27.3 kg m$^{-3}$ south of the OSNAP line. The source area is localized over the Reykjanes Ridge and the Rockall Plateau during 'small' winters (Fig. 7c), while it is localized over the interior of the Iceland Basin and the Rockall Trough during 'large' winters (Fig. 7d). The position of the isopycnal 27.3 kg m$^{-3}$ over the Reykjanes Ridge shifts north of the Charlie-Gibbs Fracture Zone (CGFZ) during 'small' winters, and south of the CGFZ during 'large' winters.




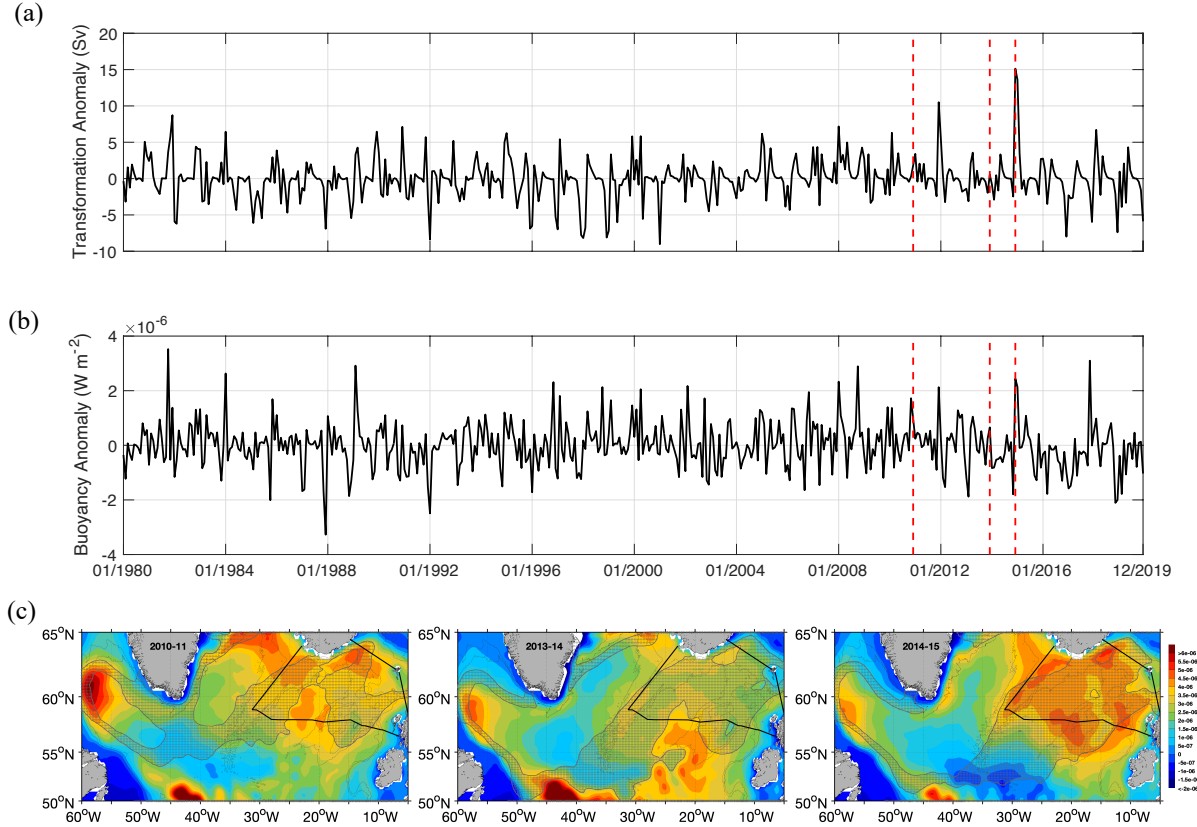

**Figure 6.** (a) Transformation anomaly at the SPMW isopycnal 27.4 kg m⁻³ between 1980 and 2019. (b) Buoyancy anomaly over the Iceland Basin between 1980 and 2019; Dashed red lines indicate transformation and buoyancy anomalies on December 2010, 2013 and 2014. (c) Maps of buoyancy forcing superimposed by the surface density area 27.3–27.5 kg m⁻³ in grey during winter (December to April) of three different years.

We explore whether the difference in these spatial patterns for the SPMW surface area can be related to an extension or contraction of the subpolar gyre. For this exploration, the subpolar gyre boundary was estimated with the largest closed contour of SSH in January of those winters (Fig. 7b). Any difference in the subpolar gyre boundary between these years is localized over the northern part of the Reykjanes Ridge and the northwest corner of the NAC. Overall, however, we cannot attribute small or large density areas to specific positions of the gyre. Similarly, the unusually large SPMW transformation in winter 2014–15 cannot be attributed to an unusually large contraction or expansion of the gyre in winter 2014–15 as compared to its position in winters 2010–11 and 2013–14 (Fig. A1).




Other possible mechanisms responsible for changes in the surface area include variability in ocean advection and the winter re-emergence of relatively dense SPMW. South of OSNAP East, an exceptionally strong heat loss (Josey et al., 2018) formed an unusually large volume of dense SPMW (Grist et al., 2016) in the winter 2013–14. Grist et al. (2016) analyzed the long-term impact of anomalous

SPMW formation on regional climate and found that some of this water mass emerges the following autumn/winter. Thus, we suggest that the large formation of SPMW in winter 2013–14 over the southern part of the Iceland Basin contributed to the large source area of SPMW found over the northern part of the Iceland Basin the following winter 2014–15.

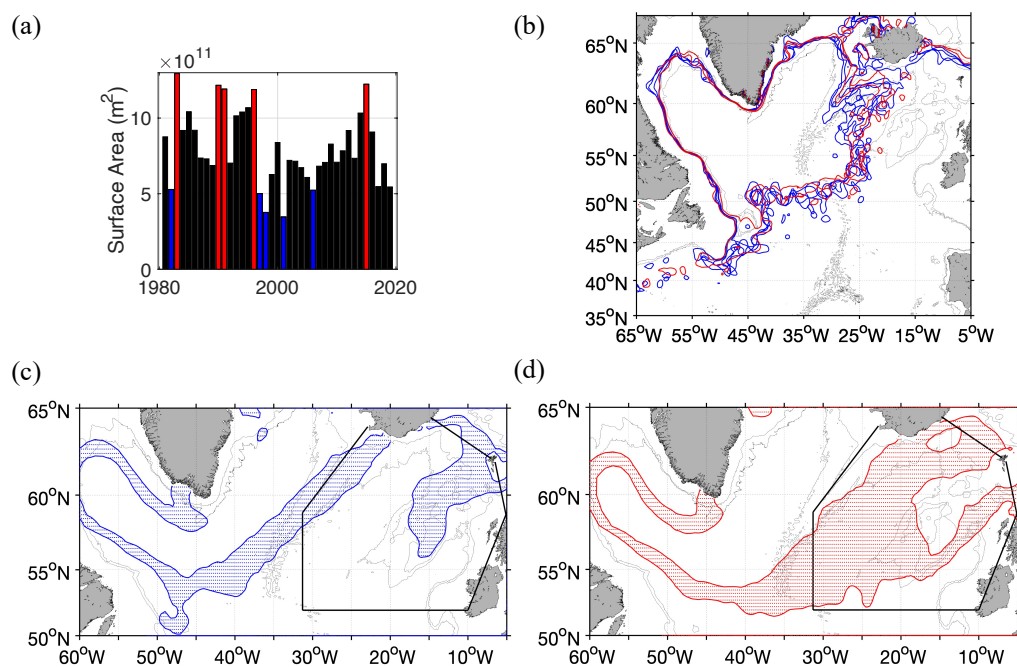


**Figure 7.** (a) Surface density area (m$^2$) in winter (December to April) for density higher than 27.3 kg m$^{-3}$ over the black box indicated in panel (c). (b) Subpolar gyre boundaries in January during years indicated by the color code. (c)-(d) Surface density area 27.3–27.5 kg m$^{-3}$ averaged during winter (December to April) of the years indicated by the color code.





## 6 Discussion and Conclusion

Using 40 years of observations (1980–2019), we show that the production of SPMW is correlated to the surface area of the source water and to buoyancy forcing over the Iceland Basin, at a similar level for each variable (R=0.7 and R=0.67 respectively). Our analysis reveals that these two variables are weakly dependent in this region: variance in the air-sea fluxes can explain ~30% of changes in the wintertime surface density field over the Iceland Basin. We thus infer that ocean advection, mixing and wind-driven

upwelling account for the remaining variability in the surface area of the source water.

The direct and indirect (via surface area changes) influences of buoyancy forcing combine to account for about half of the SPMW transformation variability. However, observation-based sensitivity experiments reveal that the spatial distribution of the SPMW transformation is most sensitive to surface density

changes, which sets the surface area for the source waters, as opposed to the direct influence of the air-sea fluxes, which is weak at these densities.

A combination of strong buoyancy forcing and a large source area during the 2014–2015 winter produced the largest SPMW transformation in 40 years. Possible causes for the large source area observed over the

northeastern part of the Iceland Basin during that winter include a large local surface heat loss during this winter, the re-emergence of a large volume of SPMW formed upstream the previous winter (Grist et al., 2016), and/or the advection of anomalously dense waters into the region.

Further exploration is needed before a specific attribution for the large transformation in winter 2014–

2015 is made, however we do note that a number of recent papers have documented strong advective changes in the Iceland Basin over the past few years. In one study (Ortega et al., 2020) the NAC is shown to drive large salinity and temperature property changes at the surface and subsurface, while in another study (Holliday et al., 2020), strong property changes are attributed to changes in the fraction of water from the Labrador Sea that reaches the Iceland Basin. These studies, highlighting the importance of ocean

advection, are consistent with a previous paper that showed large hydrographic property changes for



SPMW localized over the Reykjanes Ridge that cannot only be explained by the variability of the local air-sea fluxes (Thierry et al., 2008).

Finally, we now understand that the preconditioning of the surface waters in the Iceland Basin is a major
contributor of the total waters carried within the lower limb of the AMOC (Petit et al. 2020), and thus a key determinant of AMOC variability within the subpolar North Atlantic. Hence, our study highlights the importance of understanding the factors that influence the surface density field in the Iceland Basin—whether by advection from the subtropics or the western subpolar gyre, or by the influence of local winds that bring cold water to the surface via mixing and/or upwelling, or through surface buoyancy
loss—since that factor is of prime importance in determining the transformation of SPMW.

**Appendix A**

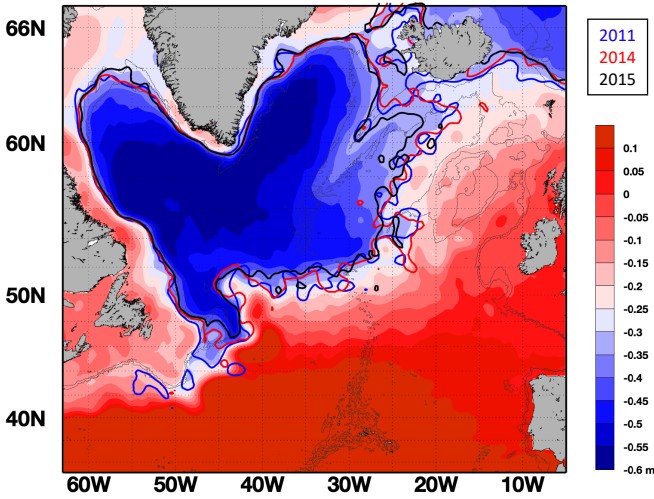

**Figure A1.** Time-mean SSH of the subpolar gyre (1993–2015) with contour interval of 5 cm. Contours in color indicate the subpolar gyre
boundaries in January 2011, 2014 and 2015 defined with the 1-cm contour intervals of the monthly SSH field.



*Data availability.* The OSNAP observations are archived online at https://www.o-snap.org/observations/data/, and the atmospheric reanalysis are accessible online at
https://www.ecmwf.int/en/forecasts/datasets/ reanalysis-datasets/era5. The subsurface salinities derived from EN4.2.1 are available at the Met Office website (https://www.metoffice.gov.uk/hadobs/en4/download-en4-2-1. html#l09_profiles).

*Author contributions.* T.P. and M.S.L. led the data analysis and S.A.J. and S.C.U. assisted with the
interpretation of the results. All the authors contributed to writing of the manuscript.

*Competing interests.* The authors declare no competing interests.

*Acknowledgements.* The T.P. and M.S.L. acknowledge the Physical Oceanography Program of the US
National Science Foundation (Code 3331843). S.A.J. acknowledges funding from the UK Natural Environment Research Council, including the North Atlantic Climate System Integrated Study program (reference number NE/N018044/1). S.C.U. was supported by UK NERC National Capability programme the Extended Ellett Line and CLASS (NE/R015953/1), and NERC Large Grant UK OSNAP (NE/K010875/1) and from the European Union's Horizon 2020 research and innovation programme
under grant agreement No 678760 (ATLAS) and No 727852 (Blue-Action).

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
