# Peer review of "Role of air-sea fluxes and ocean surface density on the production of deep waters in the eastern subpolar gyre of the North Atlantic"

_Ocean Science, 2021_

## Referee Comment (RC2)

[Figure]

**FIGURE 5.15** Annual mean air–sea buoyancy flux converted to equivalent heat fluxes (W/m²), based on Large and Yeager (2009) air–sea fluxes. Positive values indicate that the ocean is becoming less dense. Contour interval is 25 W/m². The heat and freshwater flux maps used to construct this map are in the online supplement to Chapter 5 (Figure S5.8).

From Talley, L.D., Pickard, G.L., Emery, W.J. (Eds.), 2011. Descriptive physical oceanography: an introduction, 6th ed. ed. Academic Press, Amsterdam ; Boston.

---

## Author Response (AR1)

We thank all Referees for their insightful reviews that have helped us improve the manuscript.

Dear editor and authors,

This well-written manuscript presents interesting and significant results highlighting the important role of ocean surface density variability in the eastern subpolar North Atlantic for driving changes in water mass transformation. I recommend publishing the manuscript subject to minor revisions.

Detailed comments:

Title: Consider changing the preposition in the title. For example, you could say "Role of air-sea fluxes and ocean surface density for the production…" or "Role of air-sea fluxes and ocean surface density in the production..." Alternatively, you can use "Impact of air-sea fluxes and ocean surface density on the production…"

We changed the preposition, and the title now reads:

"Role of air-sea fluxes and ocean surface density in the production of deep waters in the eastern subpolar gyre of the North Atlantic."

Line 69, "density": Do you mean potential density referenced to the surface? (Here and at several other instances throughout the text.)

We used potential density referenced to the surface to analyse the influence of the surface density field on the transformation of surface water. This is now clarified in the caption of Figure 1b and in section 2.2 (l.127-128):

"Temperature and salinity are used to calculate potential density referenced to the surface, $\sigma_\theta$ (Gill, 1982)."

Line 116, "is the potential density": Do you mean the density referenced to each vertical level z? You are computing vertical stratification, so I imagine that in this particular case you are not using potential density referenced to the surface.

You are right, the equation is now clarified:

"We also use EN4.2.1 to calculate potential vorticity ( $\frac{-f}{\sigma 0} \frac{\partial \sigma}{\partial z}$ ) in the Iceland Basin, where f is the Coriolis parameter, $\sigma_0$ is the potential density referenced to the surface, and $\frac{\partial \sigma}{\partial z}$ is the vertical gradient of potential density at middepths."

Line 120: also cite Desbruyeres et al. (2019) https://doi.org/10.5194/os-15-809-2019

Done. The sentence now reads:

"Following past work (Desbruyères et al., 2019; Speer & Tziperman, 1992; Tziperman, 1986; Walin, 1982), the evaporation (E), precipitation (P), and net heat flux into the ocean (Q) are used to estimate the surface-forced transformation across an isopycnal, σ, as follows:"

Lines 197-198, "Though a strengthening of the buoyancy forcing generally leads to an expansion of the surface area.": The correlation does not necessarily mean that this is the direction of causality. Is it possible that the direction of causality is the other way around: an expansion of the surface area can drive a buoyancy flux anomaly?

Strong buoyancy forcing is likely to increase the area of dense water at the surface since buoyancy forcing produces dense water. However, it is not clear that the introduction of dense surface water (perhaps via advection) would increase the local buoyancy forcing. We argue here that the introduction of dense surface water increases *transformation,* but our assumption regarding the correlation between buoyancy flux anomalies and surface area change is that the former drives the latter.

Line 199: You could change "30%" to "less than 30%" if R2 is closer to 0.27 than 0.30 (R=0.52 according to line 209).

Thank you for the suggestion, the sentence now reads:

"Though a strengthening of the buoyancy forcing generally leads to an expansion of the surface area, the buoyancy flux in a given winter explains less than 30% of the surface density change in the Iceland Basin."

Figure 3: What does panel (f) represent, and what are the units in that panel?

Panel (f) of Figure 3 shows the same as panel (d), namely the spatial pattern of SPMW transformation variance, zoomed in over the Iceland Basin. The units, in Sverdrup, have been added in that panel.

Lines 235-247: Consider adding some discussion on whether the surface density variability is dominated by salinity or temperature. This could make a difference, as salinity variability does not directly drive local surface buoyancy fluxes, while temperature variability does.

New supplementary figures compare the spatial pattern of the variance in SPMW transformation when it is estimated with (1) climatological surface temperature and variable surface salinity, and (2) climatological surface salinity and variable surface temperature. These results show us the importance of understanding the mechanisms that drive temperature and salinity variability, particularly the latter for the Iceland Basin.

We now discuss this result in the manuscript (l.250-253):

"The variance in surface density has contributions from both temperature and salinity variability (Fig. A2), as expected in these high latitudes where surface waters are so cold."

Line 271, "the buoyancy anomaly": Do you mean "the buoyancy flux anomaly"?

Yes, the sentence is now clarified:

"Though the buoyancy flux anomaly is relatively strong during this winter, it is not the largest anomaly observed over the 40-year record (Fig. 6b), suggesting that additional conditions are required for large transformation."

Line 323: Consider changing "dependent" to "interdependent" (which sounds more physically intuitive) or "statistically dependent" (which is a more mathematical phrase).

Done. The sentence now reads:

"Our analysis reveals that these two variables are weakly interdependent in this region: variance in the air-sea fluxes can explain ~30% of changes in the wintertime surface density field over the Iceland Basin."

Supplementary Figures:

[Figure]

**Figure A2**. Variance in SPMW transformation estimated from (left) monthly variable surface salinity and climatological surface temperature; (right) monthly variable surface temperature and climatological surface salinity.

Dear editor and authors,

This interesting and well-written manuscript examines the effect of both buoyancy fluxes and the surface area of sub-polar mode water on the production of deep waters in the Iceland Basin, and makes a convincing attempt to quantify their inter-dependence. I recommend publishing the manuscript subject to minor revisions.

**General comment**

Overall, this manuscript is well-written, clear, and with few errors. I found the sensitivity experiment especially convincing visually but wondered if the authors had considered quantifying the results of the three experiments in some way?

We estimated a ratio of the variance in transformation for the experiments (2) and (3). The variance of each experiment was divided by the variance of the experiment of reference, experiment (1). The ratios at each grid point were then averaged over the Iceland Basin for the density range 27.3–27.5 kg m$^{-3}$, and over the Irminger Sea for the density range 27.5–27.6 kg m$^{-3}$. As suggested in Figure 4, we find a larger ratio of 0.92 in experiment (3) than in experiment (2), of 0.44. Similar results are obtained for the transformation of surface waters to a density > 27.55 kg m$^{-3}$ over the Irminger Sea, with a ratio of 0.91 for experiment (3) and 0.41 for experiment (2).

We now discuss these estimations at l.243-245:

"Over the Iceland Basin, we estimate a ratio of the variance in SPMW transformation over the total variance of 0.44 for the experiment (2) and 0.92 for the experiment (3). […] Similar results are obtained for the transformation of surface waters to a density $\sigma_\theta$ > 27.55 kg m$^{-3}$, which is the averaged isopycnal of the maximum AMOC at OSNAP East (Fig. 4, lower row), with a ratio of 0.41 for the experiment (2) and 0.91 for the experiment (3) over the Irminger Sea."

I found the figures generally very good and clear, although some colourbar labels were very small.

Colorbar labels are now bigger in all the figures.

**Major comments**

Figure 3: The captions and figures for (a) and (b) are mismatched - (a) is labelled Area on the x-axis but captioned buoyancy flux, and vice versa.

Thank you for pointing out the mismatch, the caption of Figure 3(a,b) now reads:

"Correlations between the SPMW transformation to densities higher than 27.4 kg m$^{-3}$ in the Iceland Basin and the (a) surface area of 27.3–27.5 kg m$^{-3}$ in the Iceland Basin, and (b) buoyancy flux over the surface area of 27.3–27.5 kg m$^{-3}$ in the Iceland Basin. The dependence between the surface area and the buoyancy flux was removed to compute

their correlations with the SPMW transformation. All correlations are computed using winter values (December to April)."

**Minor comments**

Figure 1(a): Labelling the isopycnals would help with identification. I assume that the darker shading is for greater sea surface density, but a greyscale colourbar would also help. Also, I assume that the black lines define the study domain described in lines 150--152 but stating this would be useful.

The isopycnals are now labelled in Figure 1a and the greyscale and domain of study have been better described in the caption, which now reads:

"Sea surface density (kg m$^{-3}$) averaged over winter (December to April) 1980–2019; contour interval is 0.1 kg m$^{-3}$. Dark grey shows dense surface water of $\sigma_\theta > 27.7$ kg m$^{-3}$ in the Irminger Sea. White dashed lines outline the isopycnals 27.3 and 27.5 kg m$^{-3}$. The OSNAP East section, divided into 7 coloured subsections, forms the southern boundary of our closed domain. The northern boundary is indicated by black lines."

Figure 1(b): This is a nice, clear plot and the descriptive caption and colour matching with Fig. 1(a) makes it very easy to understand.

Figure 1(c):  A little more description in the caption would be helpful, e.g, 'Surface area (m$^2$) between the 27.3 and 27.5 kg m^-3 isopycnals over the Iceland Basin in January'

Done.

Lines 150--160: I'd be interested to know what the area of the study domain is, and how the surface area varies as a percentage of this domain.

The total area of the study domain is 9.13 10$^{11}$ m$^2$. Thus, the mean surface area of the source water represents 40.2% of the total area and varies by 16.8%. It is now indicated in lines 150-160:

"The mean surface area of the source water over the Iceland Basin is 3.67x10$^{11}$ m$^2$ (40.2% of the total area in the study domain) and is highly variable over the period 1980–2019, with a standard deviation of 1.53x10$^{11}$ m$^2$ (Fig. 1c)."

Lines 164--166: Is the gyre boundary defined as the largest closed contour within a defined region?

We defined the gyre boundary as the largest closed contour over the entire subpolar gyre, without defining a specific region. The sentence is now clarified:

"Following previous work, the gyre boundary is defined as the largest closed contour of the monthly SSH field with 1-cm contour intervals over the subpolar gyre (Foukal & Lozier, 2017)."

Figure 2 (a):

- Isn't W m$^{-2}$ a unit of heat flux rather than buoyancy flux? I appreciate that it is common practice to map buoyancy flux to heat flux (see Figure 5.15 from Talley et al., 2011, see attached PDF) but if this is the case could it be stated explicitly.
- shouldn't positive heat/buoyancy flux lead to density loss (see Figure 5.15 caption)?
- The magnitudes of the buoyancy/heat flux seem very small, compared to those shown on this figure.

I understand what the plot is showing from the descriptions in the text, but I'm somewhat confused by the plot itself. Some clarification would be appreciated.

Contrary to Talley et al. (2011), the buoyancy flux in Figure 2a is estimated following equation 1, the term in square brackets, and thus is in W m$^{-2}$. We apply the thermal expansion coefficient $\alpha$ to the net heat flux and invert the signs so that a positive buoyancy flux leads to a densification of the surface water. The term is then integrated over the surface area of a density bin to estimate transformation across the associated isopycnals, with positive transformation leading to a densification of the surface water.

We clarified the caption of Figure 2a, which now reads:

"Figure 2. (a) Buoyancy flux (W m$^{-2}$) and (b) potential vorticity (m$^{-1}$ s$^{-1}$) averaged over the source density area in the Iceland Basin. Positive buoyancy flux leads to a densification of the surface water (term in square brackets in Equation 1). The isopycnals 27.3 and 27.5 kg m$^{-3}$ do not outcrop over the Iceland Basin during summer. The dashed black line shows the potential vorticity 4x10$^{-11}$ m$^{-1}$ s$^{-1}$."

Lines 174--179: Saying buoyancy loss/gain rather than flux would be clearer.

Done. The sentence now reads:

"The 2014–2015 winter stands out among these profiles, as it is marked by both the strongest buoyancy loss (8x10$^{-6}$ W m$^{-2}$ in December; Fig. 2a) and the deepest SPMW (600 m in March; Fig. 2b). Conversely, the 2016–2017 winter is associated with a weak buoyancy loss (4x10$^{-6}$ W m$^{-2}$ in December) and a shallow SPMW (250 m in March)."

Lines 177--201: The approach here is convincing, comparing both SPMW thickness and surface area to buoyancy forcing. I think stating an explained variance of less than 30% is fine and would strengthen the argument further, given that $R^2 = 0.27$.

Done. The sentence now reads:

"Though a strengthening of the buoyancy forcing generally leads to an expansion of the surface area, the buoyancy flux in a given winter explains less than 30% of the surface density change in the Iceland Basin."

Lines 201--203: It could be argued that 27% is sufficient contribution by buoyancy flux to surface density changes for them not to be regarded as independent, but I think the sensitivity experiments address this satisfactorily.

Figure 3: The (c-e) and (d-f) labels are a bit misleading, as (c-e) looks like (c, d, e) to me. I would prefer to see (c, e) and (d, f). Are these the same plots just zoomed in on the study domain and with slightly different scales? It might be helpful for the caption to say so.

Done. The caption now reads:

"(c, e) Variance in buoyancy flux (W m$^{-2}$) in winter, with (e) a zoom of (c) over the study domain. (d, f) Distribution of the interannual variability in SPMW transformation (Sv), with (f) a zoom of (d) over the study domain."

Line 227: The almost inverse visual relationship of buoyancy flux and surface area variability within the study domain (Fig. 3 (e) and (f)) is interesting and supports the conclusion in 232--234.

Line 335: Isn't 'a large surface heat loss' another way of saying a large buoyancy flux? Would this suggest investigating the relationship between the buoyancy flux of the previous winter with the SPMW transformation of the current winter (i.e., a one-year lag) as an additional contributor?

Indeed, an interesting follow-up study could investigate the indirect effects of buoyancy flux by considering a one-year lag for its local effects (remnant dense water formed by large buoyancy flux inside the study area during previous winters) and its remote effects (the advection of dense water formed by large buoyancy flux outside the study area during previous winters).

Yours sincerely,

Emma Worthington

**References**:

Talley, L.D., Pickard, G.L., Emery, W.J. (Eds.), 2011. Descriptive physical oceanography: an introduction, 6th ed. ed. Academic Press, Amsterdamâ ¯; Boston.

Review of the manuscript "Role of air-sea fluxes and ocean surface density on the production of deep waters in the eastern subpolar gyre of the North Atlantic" by Petit et al.

This is a concise piece of work on the important aspect of large-scale water mass transformation. However, I find the apparent implications to AMOC and climate weak. Also, I have issues with some of the analyses and related statements. I thus find the manuscript in need of a major revision. I hope the authors find my comments useful in doing so.

GENERAL COMMENTS

AMOC/AMOC-link: inference/assertions related to AMOC are relative prominent in the MS, particularly in the abstract and conclusion. But any direct link, at least explicitly - and particularly related to variability, is absent from the study. Actual AMOC (co-)variability must be added or the AMOC implications must be much less prominent and anyway presented as no more causal than the study warrants. What is the relation of, e.g., Fig 6a, to a relevant AMOC time series?

The study analyzes the factors that drive variability in the production of Subpolar Mode Water (SPMW) in the Iceland Basin (Figure 6a). Past studies have demonstrated that SPMW is a major contributor to the total waters carried within the lower AMOC limb (Chafik & Rossby, 2019; Lozier et al., 2019; Petit et al., 2020) and that its formation pre-conditions the formation of dense water further downstream (de Boisséson et al., 2012; Brambilla et al., 2008). In this paper, we reveal the key role of the surface density field in the production of SPMW. This understanding of the mechanism that drives the variability of SMPW will aid predictions of AMOC variability in the years and decades ahead. For these reasons, the link between SPMW transformation and AMOC is discussed in the manuscript.

While we believe that the link between AMOC and SPMW transformation is sufficiently and clearly discussed in the conclusion, we have modified the abstract for clarity.  Lines 16-17 now read: "We analyze these contributions to the transformation in order to better understand the connection between atmospheric forcing and the densification of surface water."

strength of (co)relations: there is a use of qualitative statements, e.g., "weak" (l.178), "strong" (l.206), "weakly dependent" (l.323) that are unjustified by the specific numbers presented. (r=0.52 is quite substantial in real interannual data; if the former is "weak", r=0.67 is not "strong"). Relatedly, there is no reference, not to say quantification, of significance, confidence levels nor confidence intervals

We now indicate the significance of each correlation throughout the manuscript by adding the associated p-values:

l.192: "All correlations have p-values < 0.05."

l.207: "(0.66 and 0.63, respectively, with p-values < 0.05)").

The interpretation of those correlations is also clarified in the text, as indicated below.

Figures: several of the panels are a little too stamp-sized as they appear; some cumbersome units/crowded axes (e.g., 2c

Panels in Figures 4 and 5 are now larger and the color bar labels are bigger in all the figures. The axes in Figure 2(c) have been detailed for a clear comparison with Figure 2(d).

papers to add?

From the top of my head, here's a few papers (non-exhaustive list) the authors could benefit from relating to in revising their manuscript:

Johnson et al (2019). Recent contributions of theory to our understanding of the Atlantic Meridional Overturning Circulation. Journal of Geophysical Research: Oceans 12 (8), 5376–5399.

Langehaug et al (2012) Water mass transformation and the North Atlantic Current in three multi-century climate model simulations. J. Geophys. Res., 117, C11001

We now refer to Langehaug et al (2012) in the introduction and to Johnson et al (2019) in the conclusion. We also cite Desbruyere et al. (2019) in section 2.

Desbruyères, D. G., Mercier, H., Maze, G., & Daniault, N. (2019). Surface predictor of overturning circulation and heat content change in the subpolar North Atlantic. *Ocean Science*, *15*(3), 809–817. https://doi.org/10.5194/os-15-809-2019

SPECIFIC COMMENTS

l.10: "convection", is this the specific term you want to use? First, this seems the only place its used in the MS, and, second, at least to this reviewer, it gives the association to outdated an understanding of AMOC etc via specific convective sites that

The term "wintertime convection" refers here to the process of light water densification that forms the lower limb of the MOC at high latitudes, not to specific convective sites.

l.22–23, "... is set by ... advection, wind-driven upwelling...": neither are addressed herein?

We clarified the sentence that now reads:

"This surface density is set by a combination of advection, wind-driven upwelling and surface fluxes. Our study shows that the latter explains ~30% of the variance in outcrop area as expressed by the surface area between the outcropped SPMW isopycnals."

l.31–34, interannual variance in AMOC rooted in subpolar NA rather than in the Nordic Seas: What is the evidence/refs for this. What meassure of dense water production and AMOC does the statement relate to?

The statement relates to measures of overflow transports along the Greenland-Scotland Ridge from Bringedal et al (2018), and to their comparison with the AMOC variability at OSNAP East by Petit et al. (2020). The references have been modified accordingly.

l.39: awkward combination of range(?) and uncertainty/interannual(?) variance

The average overturning has been shown by Petit et al. (2020) and now reads "6.6 ± 3.8 –7.6 ± 3.8 Sv" in the manuscript. The range of 6.6–7.6 Sv is explained by the two estimations of volume budget for the upper and lower layers between the Greenland-Scotland Ridge and OSNAP East; ± 3.8 Sv indicates the error associated with both volume budgets, as explained in their section 2.

l.40, "drives AMOC variability": assertion, please specify in what sense, by what evidence

The assertion refers to the interannual variability over the OSNAP period (2014-2016), as shown by Petit et al. (2020), which is cited at the end of the sentence. A striking correspondence between the low-pass filtered variability of the AMOC and transformation rates has also been shown by Desbruyeres et al. (2019).

l.53–64: here you raise several interesting aspects regarding how AMOC varies and what may (not) cause it. However, I cannot see that you return to these aspects at all. E.g., to what extent does your study add to resolving the issue(s)? Regarding what measure of AMOC?

This paragraph summarizes past work that has shown large property changes in the subpolar gyre over the past several years. Our study investigates the role of these property changes on the transformation of surface water, by focusing on SPMW densification. As indicated above, the densification of SPMW is a key process in the pre-conditioning of dense water formation downstream (de Boisséson et al., 2012; Brambilla et al., 2008).

l.65–73, fixed PV and density-class boundaries: how adequate are such fixed classifications in a variable/warming climate? And in general, AMOC is not fixed to overturn within a fixed density "window"? (general question, by no means specific to this study)

We use fixed density boundaries because we are interested in the evolution of this specific layer over 40 years, which is called SPMW over the Iceland Basin. More precisely, we include various class of SPMW by considering a large density window, thus we are less impacted by the climate change at longer time scales.

At the time of the study, the data provided by the OSNAP program were only available until May 2018 (Li et al., 2021).  OSNAP data from 2018 to 2020 are still being processed.

l.110,113: are ERA5 and EN4.2.1 consistent/compatible data sets? Could you have settled for using only one? And how about AVISO in defining the gyre/area?

The ERA5 atmospheric reanalysis does not provide salinity at the surface, and EN4.2.1 does not provide air-sea fluxes. Thus, we combine these two data sets to estimate the transformation. Unfortunately, there is no observational dataset that provides both air-sea fluxes and hydrological properties at the sea surface. Sea surface height from AVISO was used to define the gyre boundary following Foukal & Lozier (2017).

The time range for the AVISO data has been added, and the sentence now reads:

"To estimate the subpolar gyre boundary, we use monthly absolute dynamic topography fields from the gridded ¼° AVISO (Archiving, Validation and Interpretation of Satellite Oceanographic data center) altimeter products distributed by CMEMS (Copernicus Marine Environment Monitoring Service) in 1993–2019."

l.177-179: I disagree with this qualification, in my experience this is quite a substantial correlation in climate-related data. Relatedly – "lower than expected", what was expected? And how about the compatibility between the two data sources used to diagnose the respective fields?

We agree that a correlation of 0.52 is non negligible in climate-related data, but we can consider these variables as weakly correlated because they explain less than 50% of the variance between the buoyancy flux and the change in SPMW thickness, in opposition to the correlation of more than 0.6 found between the buoyancy flux and the SPMW transformation in Figures 3(a), which explains more than 50% of their variance. The sentence is now clarified:

"Despite these agreements, the linkage between the SPMW thickness and the buoyancy flux over the period 1999-2019 is relatively weak (< 50% of their variance), with a correlation of 0.52 (Fig. 2c)."

According to the suggestion from Reviewer 1 and Reviewer 2, we changed "only 30%" to "less than 30%" because R2 is closer to 0.27 than 0.30 (R = 0.52), and the sentence now reads:

"Though a strengthening of the buoyancy forcing generally leads to an expansion of the surface area, the buoyancy flux in a given winter explains less than 30% of the surface density change in the Iceland Basin."

l.202–204, "sufficiently small... so that we can separately explore": I disagree, follows from the above. Use regression or similar to remove covariance and do statistics on what remain. Now it appears to be assertion in the place of analysis.

Thank you for the suggestion, we estimated a linear regression between the surface area and the buoyancy flux, and removed it from their correlations with the SPMW transformation in Figure 3(a) and (b). The correlations are now of 0.63 and 0.67, respectively. The sentence in l.202-204 now reads:

"To conclude, we removed the weak dependence of the surface density field on the air-sea fluxes so that we can separately explore the contributions of the air-sea fluxes and ocean surface densities on the interannual variability of SPMW transformation."

And the caption in Figure 3(a) and (b) now reads:

"Correlations between the SPMW transformation to densities higher than 27.4 kg m$^{-3}$ in the Iceland Basin and the (a) surface area of 27.3–27.5 kg m$^{-3}$ in the Iceland Basin, and (b) buoyancy flux over the surface area of 27.3–27.5 kg m$^{-3}$ in the Iceland Basin. The dependence between the surface area and the buoyancy flux was removed to compute their correlations with the SPMW transformation. All correlations are computed using winter values (December to April)."

l.236, "observation-based experiments": is "experiment" the most appropriate term here? Isn't it more like a diagnostic/analysis?

Done. We now use the term "sensitivity analyses".

l.306–307, advection and re-emergence: this is a possibility, but what's the evidence? Can you add some independent analyses or at least some robust arguments?

We have now clarified that the possibility of re-emergence comes from the work of Grist et al. (2016). The sentence now reads:

"Thus, it is possible that the large formation of SPMW in winter 2013–14 over the southern part of the Iceland Basin contributed to the large source area of SPMW found over the northern part of the Iceland Basin the following winter 2014–15."

l.309–310, "impact ... on regional climate". This is unsubstantiated as it stands. And even if the Grist et al-paper referred to alludes to it, I don't find it substantiated there either.

Grist et al (2016) provides "evidence for the re-emergence of anomalously cold SPMW in early winter 2014/2015" by generating ensembles of particle trajectories in ORCA12 (their Figure 16). Thus, they assess the regional impact of extreme air-sea interactions over the eastern subpolar gyre.

l.311–313, "Thus...": is there a larger/AMOC/climatic implication to be substantiated here?

The implication here discusses the possibly strong indirect effect of buoyancy flux on the SPMW transformation (e.g., the advection of dense water formed by large buoyancy fluxes outside the study area during previous winters). An investigation of the impact of these events on the AMOC could be an interesting follow-up study, but it is out the scope of our analysis.

Section 5, here and elsewhere: I sometimes miss an explicit reference to the quantifications done. Does this refer to the (non-numbered) Eq 1?

Equation 1 is now numbered in the manuscript and referenced in section 5 (see answer below). The caption in Figure 6(a) now reads:

"Figure 6. (a) Transformation anomaly at the SPMW isopycnal 27.4 kg m$^{-3}$ between 1980 and 2019, as estimated in Equation (1)."

Fig. 6 and its description/quantification: where does the Sv come from (see above comment)? And how can the reader translate/compare this to eventual AMOC (or similar) anomalies? And what AMOC (or similar) anomalies would those be? None are provided in the manuscript as far as I can see.

Figure 6(a) shows the transformation anomaly at the SPMW isopycnal, as estimated in equation 1. Following this equation, the buoyancy flux (term in square brackets) is integrated over the surface area of a density bin, which is associated with the SPMW density-range. The transformation is then in Sverdrups (Desbruyères et al., 2019).

The SPMW transformation over the Iceland Basin cannot be directly compared to AMOC anomalies at OSNAP East because the transformation is estimated at a lighter density than the AMOC isopycnal and does not include deep waters formed in the Irminger Sea. Moreover, we cannot directly compare the transformation over 40 years with the AMOC anomaly at OSNAP East as the AMOC measurements at the OSNAP line are available over 4 years, which highlights the necessity to obtain longer direct measurements.

l.320, 40 years of observations: I apologise, as this may reveal my sloppy reading. But the observational focus is still not perfectly clear to me by now – is it 1980–2019 and/or 2014–2018. This could maybe be even clearer, eg in the introduction.

Our study analyses the transformation of SPMW over the 40 years from 1980 to 2019, as indicated in the introduction (l.84-85).

l.322–323: same as above, downplaying what to me seems substantial correlations

See the answer above.

l.324–325, "We thus infer": same as above, inference as a more or less reasonable statement, but with no real analyses or substantial reasoning offered.

We clarified the sentence that now reads:

"We thus infer that other mechanisms influencing density changes, including ocean advection, mixing and wind-driven upwelling (Johnson et al., 2019), account for the remaining variability in the surface area of the source water."

l.336, and elsewehere, "re-emergence": it's not clear to this reviewer what is specifically referred to, nor how the authors arrives at this conclusion/suggestion

See the answer above. Re-emergence refers to Figure 16 of Grist et al. (2016).

l.342: NAC had been found literally "to drive" or rather "to advect" anomalies?

Done. The sentence now reads:

"The NAC is shown to advect large salinity and temperature property changes at the surface and subsurface (Holliday et al., 2015), with strong property changes attributed to changes in the fraction of water from the Labrador Sea that reaches the Iceland Basin (Holliday et al., 2020)."

Fig 1: you may want to zoom out a little in Fig 1a to provide more regional context (and a convenient intro illustration)

Done.

Fig 6, caption "during years indicated by color code": too implicit/subtle

The caption in Figure 7 has been clarified and now reads:

"(b) Blue (red) contours show the subpolar gyre boundaries in January during winters of 'small' ('large') area, as identified in panel (a). (c)-(d) Surface density area 27.3–27.5 kg m$^{-3}$ averaged during winters of 'small' (blue) and 'large' (red) area, as identified in panel (a)."